# Wide pH Range Potentiometric and Spectrophotometric Investigation into the Acidic Constants of Quercetin, Luteolin and L-Ascorbic Acid in Aqueous Media

**Luana Malacaria and Emilia Furia ***

Department of Chemistry and Chemical Technologies, University of Calabria, 87036 Rende, Italy
* Correspondence: emilia.furia@unical.it; Tel.: +39-0984-492831

**Abstract:** It is now well established that the dissociation constants of an organic compound are characteristic of the types of groups, or the combinations of groups, contained in it. Furthermore, the acid–base dissociation constants are important parameters to fully understand the properties of a molecule in biological systems. In this framework, the aim of the present study was to determine the acidic constants of three natural molecules with well-known antioxidant properties, namely quercetin, luteolin and L-ascorbic acid. The evaluation was carried out in aqueous media (i.e., 0.16 M NaCl) at 37 °C in a wide pH range by using a combined approach based on potentiometric and spectrophotometric measurements. The results underline the necessity to employ both experimental techniques to obtain accurate values for acidic constants, preventing uncertainties related to undesirable oxidation reactions.

**Keywords:** quercetin; luteolin; ascorbic acid; acidic constants; potentiometry; UV-vis spectrophotometry

## 1. Introduction

The thermodynamic properties of bioactive molecules in aqueous solutions are of significant scientific and practical interest. In this context, the acidic constants are important physicochemical parameters of a molecule, as they reflect its reactivity, solubility, affinity for a macromolecule and/or ability to establish intermolecular interactions. In addition, the acid and basic functional groups of a molecule strongly influence its pharmacokinetics and toxicity. The experimental evaluation of the p$K_a$ values of the molecules of biological interest, which could be hygroscopic and very unstable in an extreme pH range, is still unsolved, as it is related to the conditions in which the experiments are performed. The most useful and widely employed conventional methods are potentiometric and spectrophotometric titrations and conductometry [1–3]. However, only a few of them have referred to measurements conducted in aqueous media, where they show low water solubility.

In this framework, the present study intends to evaluate the acidic constant values of quercetin, luteolin and L-ascorbic acid (Figure 1a–c), three natural antioxidants (generically $H_nL$) whose beneficial properties are well known, in aqueous solution. The experimental approach provides measurements via potentiometric and UV-vis spectrophotometric titrations in a wide pH range, at 37 °C and ionic strengths of 0.16 M, using NaCl as a background electrolyte. The constant ionic medium method has proved to be indispensable in equilibrium studies of complex ionic reactions as it is necessary to minimize activity coefficient variation, even though the modification is in the concentration of the reagent. The method, which consists of using concentrated aqueous solutions of inert salts as the solvent, allows for the use of concentrations instead of activities in the evaluation of equilibrium constants. Additionally, it was useful to minimize the liquid junction potential, owing to the hydrogen ion concentration, which was varied in a wide pH range to assess the acidic constants of $H_nL$.

**Figure 1.** Chemical structure of the investigated natural antioxidants: quercetin (**a**), luteolin (**b**) and L-ascorbic acid (**c**).

Quercetin and luteolin belong to flavonoids that are secondary plant metabolites of the phenolic family. The most common flavonoids contain three rings, indicated as A, B and C (Figure 1a,b), and the oxidation state of C ring provides their classification into several subgroups, as quercetin and luteolin belong to flavonol and flavone subclasses, respectively. L-ascorbic acid is a water-soluble vitamin required for the prevention of scurvy and is known to be active as a cellular hydrogen transfer carrier for the redox enzyme in live cells [4]. Structurally, ascorbic acid is a γ-lactone and an ene-diol (Figure 1c).

All these plant-derived natural compounds are found in significantly high quantities in fruits and vegetables and are considered valuable health-promoting compounds [4–9]. Choosing these natural antioxidants is also related to their well-known capability to chelate several metal ions of relevant importance for biological systems, and they have attracted our research interest in the past 10 years [10–17]. Their ability to bind cations depends on their structure, characterized by functional groups such as phenol, enol and enediol moieties that can undergo deprotonation via metal interaction, whatever their p$K_a$ value. In this context, the accurate knowledge of the acidity of -OH is fundamental to understanding how these molecules can counteract the metal-related damage [18–25]. Furthermore, their antioxidant ability could be correlated to the relative acidity of -OH groups.

## 2. Materials and Methods

### 2.1. Chemicals

All chemicals were of analytical grade. The titrant carbonate-free solutions of sodium hydroxide, the hydrochloric acid and the sodium chloride stock solutions were prepared and standardized as previously described [12].

Quercetin (Sigma, ≥95%), luteolin (Sigma, ≥98%) and L-ascorbic acid (Sigma, 99.2%), were stored in a desiccator over silica gel and used as such.

To avoid variations during dilution, all the solutions were prepared by adding the appropriate amount of NaCl, as the background electrolyte, to set the ionic strength at 0.16 M. Double-distilled water, freed from any organic impurities by means of a Milli-Q system (Millipore), was used to prepare fresh solutions.

### 2.2. Potentiometry and Spectrophotometry

The potentiometric arrangement and the titrations were conducted with the same apparatus described previously [12]. The glass electrodes were from Metrohm. Following the addition of the reagents, they acquired a constant potential (within ±0.01 mV) by 30 min. To prevent any interference due to carbonate formation, nitrogen gas was gently introduced into the test solutions, which were magnetically stirred throughout titrations.

Nitrogen commercial cylinders were used, and the gas was further purified by passing it through 1 M NaOH, 1 M $H_2SO_4$, double-distilled water and 0.16 M NaCl. The cell apparatus was kept in a thermostatic bath at $(37.0 \pm 0.1)$ °C.

The acid–base equilibria were investigated by performing potentiometric titrations, at 37 °C and in 0.16 M NaCl, and by measuring, with a glass electrode (GE), the electromotive force (EMF) of cell:

$$RE|Test\ Solution|GE$$

where RE corresponds to the reference electrode (Ag | AgCl | 0.01 M NaCl | 0.16 M NaCl), while the test solution was made of $C_L$ M of $H_nL$, $C_A$ M HCl, $C_B$ M NaOH and 0.16 M NaCl. The primary $C_L$, $C_A$, $C_B$ and $[H^+]$ represent the basic data in the treatment to obtain the acidic constants.

The EMF of the cell can be expressed, in mV, at the temperature of 37 °C, as Equation (1):

$$E = E° + 61.54 \log[H^+] + E_j \tag{1}$$

where $E°$ was constant for each series of measurements and $E_j$ is the liquid junction potential, which depends on $[H^+]$ only [26]. The $E_j$ value at 37 °C and 0.16 M NaCl was determined by acid–base titration using cell (G) without $H_nL$. The obtained results for $[H^+] \leq 0.100$ M could be accurately fitted by the linear slope $E_j$ $([H^+])I = -j\ I\ [H^+]$. The value of $j$ (mV/M) was $631 \pm 1$.

All titrations were separated into two parts. In the first one, $E°$ constant value (within $\pm 0.05$ mV), was determined in accordance with the Gran's method [27,28] in the $[H^+]$ range $10^{-4}$–$10^{-2}$ M and without $H_nL$. $[H^+]$ was stepwise reduced by coulometrically generating $OH^-$ ions with the circuit C:

$$- Pt/Test\ Solution/AE +$$

where AE is the auxiliary electrode of generical composition: 0.16 M NaCl/0.1 M NaCl, (0.16–0.1) M NaCl/$Hg_2Cl_2$/Hg.

Under the chosen experimental condition, the only electroreducible species was water. Hence, suppose that at the cathode the only reaction that occurs is the water reduction (Equation (2)):

$$H_2O + e^- \rightleftarrows ½\ H_2 + OH^- \tag{2}$$

In the test solution of a known volume, $V$, $C_B$ can be expressed as $(\mu F \cdot 10^{-6}/V)$ M, where $\mu F$ stands for the microfaradays passed through the cell.

In the second part, after the addition of $H_nL$, the acidity was gradually decreased by adding known volumes of the standard NaOH solution.

All titrations were conducted with a programmable computer-controlled data acquisition switch unit 34,970 A from Hewlett and Packard. The constant–current source of the coulometric circuit was a system DC power supply 6614 C by Hewlett and Packard. The EMF values were measured at a precision of $\pm 10^{-5}$ V using an OPA 111 low-noise precision DIFET operational amplifier.

Given the low solubility of quercetin and luteolin in water [12,14] and by using the same experimental approach tested in a previous study [29], all the titrations were carried out by adding an exactly known and weighed amount of solid in the cell apparatus. When equilibria (i.e., Equations (2) and (3)) take place, quercetin and luteolin dissolved into the aqueous medium. $C_L$ varied between 0.5 and $5 \times 10^{-3}$ M, while the hydrogen ion concentration varied from $5.0 \times 10^{-3}$ M (pH 2.3) to $1.6 \times 10^{-9}$ M (pH 8.8). The higher pH limit was imposed on the accuracy in the EMF measurements by glass electrode. For this reason, all systems were studied by UV-vis spectrophotometric titrations to investigate the more alkaline pH range (i.e., up to pH 12). $C_L$ was varied between 0.01 and $0.04 \times 10^{-3}$ M. The spectrophotometric measurements were carried out with a Varian Cary 50 Scan UV Visible Spectrophotometer, by controlling the temperature of the cell holder at

(37.0 $\pm$ 0.3) °C by using a Grant circulating water bath. Matched quartz cells of thickness 1 cm were employed.

The absorbance values, $A_\lambda$, recorded to 0.001 units between 200 and 600 nm for quercetin and luteolin, and between 200 and 340 nm for L-ascorbic acid, were measured each 2 nm to collect data for numerical elaboration by using a Hyperquad [30]. The spectra were recorded 2 minutes after mixing with a strong base. This period was thought short enough to avoid any photochemical degradation. The acquisition of data was controlled with the aid of a computer linked to the instrument.

## 3. Results

Potentiometry and spectrophotometry were used to determine the acidic constants of quercetin, luteolin and L-ascorbic acid—generically $H_nL$. To acquire accurate results, the physical integrity of the molecules and the stability at different pH values in which the study was carried out must be taken into consideration during the whole experiment. So the systems were kept under an inert atmosphere and prevented from being exposed to light radiation.

The general equilibria reported in Equations (3) and (4) fully explain the potentiometric data, treated by numerical and graphical methods:

$$H_nL + H_3O^+ \rightleftarrows H_{n+1}L^+ + H_2O \qquad \log K^* (\pm 3\sigma) \tag{3}$$

$$H_nL + H_2O \rightleftarrows H_{n-m}L^{m-} + m\,H^+ \qquad \log K_{am} (\pm 3\sigma) \tag{4}$$

These correspond to the protonation of the carbonyl moiety, according to [31], and to the protolysis of the -OH groups, respectively. The data used to determine the acidic constants, according to Equations (3) and (4), were obtained by conducting two titrations for each compound.

The interpretation of the potentiometric experimental points, in terms of species and equilibrium constants, was carried out with the least-squares computer program Superquad [32] to get the minimum of the function reported by Equation (5):

$$U = \sum (E_i^{\text{obs}} - E_i^{\text{cal}})^2 \tag{5}$$

The results are reported in Table 1.

**Table 1.** Survey of the log $K^*$ and log $K_m$ values, according to general equilibria 3 and 4, by numerical methods. The uncertainties represent 3 $\sigma$.

| | log $K^*$ | log $K_{a1}$ | log $K_{a2}$ | log $K_{a3}$ | log $K_{a4}$ | log $K_{a5}$ |
|---|---|---|---|---|---|---|
| **Quercetin** | | | Superquad | | | |
| | 2.00 $\pm$ 0.06 | $-8.29 \pm 0.03$ | $-8.61 \pm 0.01$ | $-9.5 \pm 0.1$ | $-9.7 \pm 0.3$ | $-10.4 \pm 0.3$ |
| | | | Hyperquad | | | |
| | / | $-8.5 \pm 0.6$ | $-9.0 \pm 0.3$ | $-9.5 \pm 0.6$ | $-10.0 \pm 0.3$ | $-10.5 \pm 0.9$ |
| **Luteolin** | | | Superquad | | | |
| | 2.3 $\pm$ 0.1 | $-8.29 \pm 0.03$ | $-8.6 \pm 0.1$ | $-8.8 \pm 0.3$ | $-9.3 \pm 0.3$ | |
| | | | Hyperquad | | | |
| | / | $-8.1 \pm 0.3$ | $-8.80 \pm 0.05$ | $-9.6 \pm 0.6$ | $-9.8 \pm 0.6$ | |
| | log $K^*$ | | log $K_{a1}$ | | log $K_{a2}$ | |
| **L-Ascorbic Acid** | | | Superquad | | | |
| | 1.2 $\pm$ 0.1 | - | $-3.86 \pm 0.03$ | | / | |
| | | | Hyperquad | | | |
| | 1.16 $\pm$ 0.05 | | $-3.75 \pm 0.06$ | | $-10.6 \pm 0.1$ | |

The graphical evaluation was carried out by analyzing the trend of experimental points, depicted as $Z_H$ and obtained at different $C_L$ values, with respect to the pH. $Z_H$ represents the mean number of protons released for ligand [33], and it is equal to $([H^+] - C_A + C_B + K_w/[H^+])/C_L$. The values for the ion product of water, in our experimental condition (i.e., 37 °C and 0.16 M NaCl), was taken from [34].

The results are reported in Figure A1a–c, for quercetin, luteolin and L-ascorbic acid, respectively. For different $C_L$ values, the experimental points overlap, within the limit of experimental error.

As can be seen in Figure A1, the data are fully explained by Equations (3) and (4). In particular, at pH values lower than 3, $Z_H$ tends to be $-1$ for all the investigated systems, confirming that a protonation occurs, according to Equations (3) and (4).

Furthermore, as expected, the protolysis equilibria of quercetin and luteolin occur in the alkaline range ($-\log [H+] > 8$) at the expenses of the hydroxylic groups (see Figure 1a,b). In contrast, for L-ascorbic acid, the dissociation starts at a pH higher than 3 (Figure A1c) with the protolysis of -OH in position 3 (see Figure 1c).

Accurate values for the constants of protolysis equilibria (Equation (4)) were determined for quercetin and luteolin only. By performing a back titration, whose points are depicted in Figure A1c as orange crosses, we have verified that L-ascorbic acid undergoes a degradation in an alkaline medium (i.e., pH higher than 7.5), which occurs faster than the potentiometric measurement time.

Therefore, we have performed UV-vis spectrophotometric titrations at different $C_L$ values (i.e., between 0.01 and 0.04 mM). UV-vis absorption spectrophotometry is a simple and accurate method for determining acidic constants, whose determination is based on a change in the shape and intensity of the absorption spectra with a change in the pH of the test solutions.

The region of pH close to the p$K_a$ values of corresponding acids was supposed to be optimal to obtain accurate values, in the choice of the pH range for all the compounds. Additionally, the spectral range for all compounds was designated according to the variations in the spectra at both acidic and basic solutions.

Consequently, the hydrogen ion concentration was varied by adding small and exactly known volumes of a standard NaOH solution ($C_B$), from $1.0 \times 10^{-5}$ M to $1.0 \times 10^{-12}$ M for quercetin and luteolin and from $1.0 \times 10^{-2}$ M to $1.0 \times 10^{-11}$ M for L-ascorbic acid, and the absorbance values were recorded at 93 wavelengths between 230 and 457 nm for quercetin and luteolin and at 50 wavelengths between 211 and 330 nm for L-ascorbic acid.

The typical recorded spectra are depicted in Figure A2a–c, for quercetin, luteolin and L-ascorbic acid, respectively.

The UV-vis spectra of quercetin and luteolin (Figure A2a,b, respectively) show two main absorption bands, which are comparable to signals observed for most flavonoids [12,14]: band I in the 350–400 nm range, as a result of the conjugation between the B and C ring (cinnamoyl system), and band II in the 240–270 nm range, deriving from the A-C ring (benzoyl system). Moreover, two other less-intense absorption bands at 303 and 280 nm for quercetin and at 295 and at 267 nm for luteolin were observed. The UV-vis spectrum of L-ascorbic acid exhibits a single band centered at 260 nm, in accordance with the literature [11,15].

Figure A2a shows that the maximum of band I was blue shifted (i.e., from 367 to 320 nm) by moving from pH 5.5 to pH 6.5. The absorption profile changed negligibly upon raising the pH from 6.5 to 12.0, indicating, according to [35], that the acid–base sites of quercetin were not appreciably rearranged during the dissociation in this pH range.

In contrast, for luteolin and L-ascorbic acid (Figure A2b,c) the maxima of the principal absorption bands were red shifted (i.e., from 352 to 402 nm and from 243 to 267 nm, for luteolin and L-ascorbic acid, respectively) by moving from acidic to basic pH, attesting to the formation of a new spectrally active species in the solution.

The numerical elaboration of spectrophotometric data was carried out by simulating each titration from the knowledge of the total concentrations, $C_B$ and $C_L$. The minimum

of the function, described by Equation (6), was sought by employing the Hyperquad program [30]:

$$U = \sum_i \sum_k w_k \left( A_{ik} - A_{ik}{}^c \right)^2 \tag{6}$$

The ion product of water was taken from [34] and was kept invariant.

The results are given in Table 1.

## 4. Discussion

It has been suggested in previous studies that the antioxidant ability of a series of compounds is strongly affected by pH, which influences the relative abundance of the corresponding acid–base forms [36]. Acid–base equilibria have been reported to strongly affect the UV-vis spectra of several phenolics, including flavonoids. This is observed as significant changes in both the shapes and the intensities of the bands owing to variation in the pH values [37]. The absorption and fluorescence spectra of luteolin have been reported to strongly depend on pH [38]. Therefore, the corresponding acidity constant values are an essential parameter in predicting the functional mechanism of proteins in chemistry and biochemistry.

We calculated here the acidic constants of all the possible sites of quercetin, luteolin and L-ascorbic acid. According to our results, the increase in the pH of the medium at values higher than 3 prompted the proton removal from the protonated carbonyl group (i.e., from $H_{n+1}L^+$ in Equation (3)), leading to the formation of an electroneutral compound (generically $H_nL$). In contrast, in the narrow pH range of 8–10, in an aqueous solution of quercetin and luteolin, six or five species can simultaneously exist, respectively. These ion molecular forms are in dynamic equilibrium in the aqueous solution, depending on the acidity of the medium. The experimental assignment for the constants of quercetin and luteolin could not be assessed. Although some efforts have been carried out in this way, a systematic and detailed study factoring in all the possible acid forms has not been carried out, and the computational evaluation is still debated [31,35–39]. On the basis only of their structures and, in particular, on the number of phenolic -OH groups, the existence of five and four $pK_a$ values for quercetin and luteolin, respectively, can be hypothesized. However, for quercetin, in previous works dealing with its acidity constants, sets of only two [40,41], three [41–44] and four [43] $pK_a$ values have been reported. Interestingly, in the work of Álvarez-Diduk and colleagues [36], through a combined experimental and computational approach, a complete set of constants has been reported for quercetin. In particular, the deprotonation order was supposed to be 4′, 7, 3, 3′ and 5. Luteolin is structurally analogous to quercetin—except that it has four phenolic -OH groups, as there is no -OH group on ring C. Indeed, the $K_{a1}$ and $K_{a2}$ of quercetin and those of luteolin are the same. Hence, we can suppose that the first two protolysis equilibria involve -OH groups that are less influenced by the deprotonation on the C ring. Only the first acidity constant has been experimentally measured in an aqueous medium [14], and the -OH group in position 4′ has been theoretically calculated as the most acidic site [38]. In the case of L-ascorbic acid, the monoanionic form, derived from the deprotonation of the hydroxy group linked to carbon 3 [O(3)-H] (see Figure 1c), is particularly stable thanks to the delocalization of the negative charge on the oxygen atoms at positions 1 and 3. The second deprotonation occurs at higher pH values, and the corresponding constant is difficult to measure in aqueous media [45–47].

An analysis of the acidity constants in the literature shows major variation among the published values. Furthermore, most of the reported $pK_a$ values were acquired using alcohol and water combinations; thus, there is limited knowledge on $pK_a$ values found in aqueous solutions. Therefore, the acidic constant values determined in this work are not directly comparable with the others in literature, owing to differences in the experimental conditions (Table A1)—except in [11] and [14], which have reported values that are in excellent agreement with our data.

Only a qualitative comparison can be carried out with the values obtained in other works, which reveals acceptable agreement. The difference between those values and those

determined here could be related mainly to the different temperatures rather than to the medium used.

Our results showed excellent agreement between the two experimental approaches, providing a valuable method to determine acidic constants values, in a wide range of acidity, with accuracy and in aqueous media (i.e., from pH 2 to pH 12). The acidic constants determined in this work can be considered significant values in the chemistry of these ligands, and they could be indispensable to understanding ligand behavior in biological systems. Of particular biomedical importance is the ability of these compounds to form stable complexes with biologically active metal ions. The studies in this area have focused on the enhanced curing ability in the case of vitamin C and metal ion deficiencies; on the development of therapeutic agents with potential antitumor, antibacterial, antioxidant and anti-inflammatory properties with enhanced potency with respect to the noncomplexed ligands; and on their use as synthetic models for metal containing complex biological systems [13,18,48].

**Author Contributions:** Conceptualization: E.F. Investigation: L.M. Resources: E.F. Writing—original draft: L.M. and E.F. Writing—review and editing: E.F. All authors have read and agreed to the published version of the manuscript.

**Funding:** This research received no external funding. We thank the University of Calabria.

**Institutional Review Board Statement:** Not applicable.

**Informed Consent Statement:** Not applicable.

**Data Availability Statement:** Data sharing not applicable.

**Conflicts of Interest:** The authors declare no conflict of interest.

**Appendix A**

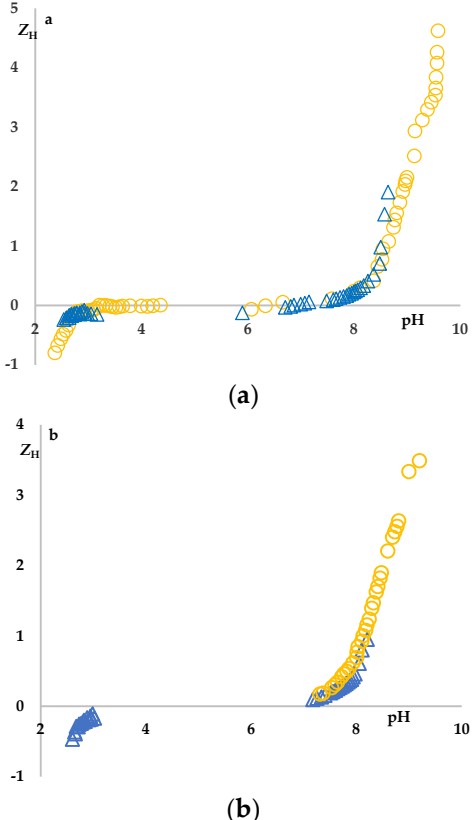

(**a**)

(**b**)

**Figure A1.** *Cont.*

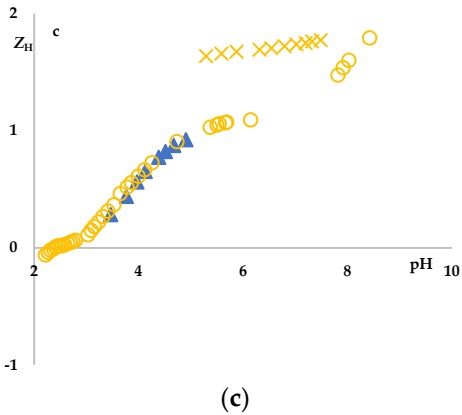

(**c**)

**Figure A1.** $Z_H$ as a function of $-\log[H^+]$ for quercetin (**a**), luteolin (**b**) and L-ascorbic acid (**c**), at 37 °C and in 0.16 M NaCl. Orange circles and orange crosses refer to $C_L$ 0.5 × 10$^{-3}$ M, while blue triangles refer to $C_L$ 5 × 10$^{-3}$ M.

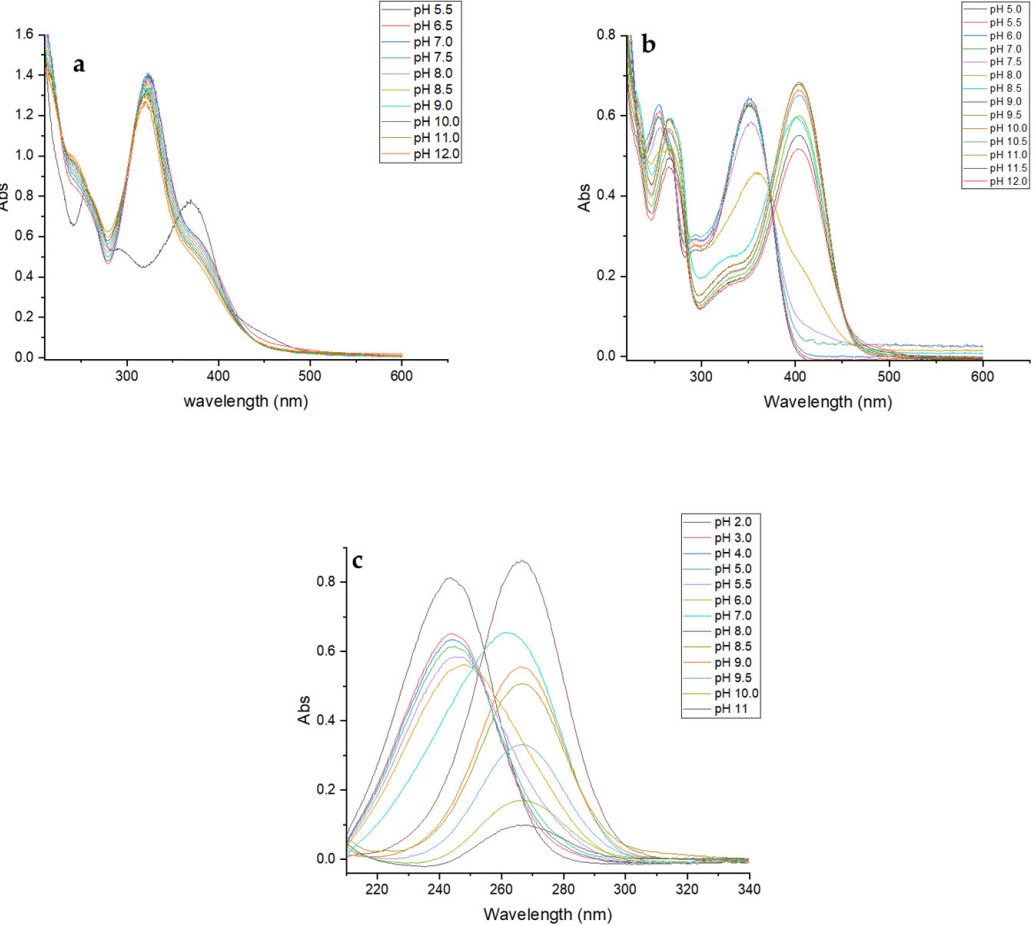

**Figure A2.** UV-vis spectra recorded for quercetin 0.025 mM (**a**), luteolin 0.010 mM (**b**) and L-ascorbic acid 0.040 mM (**c**), at 37 °C and in 0.16 M NaCl.

**Table A1.** Acidic constants of quercetin (Que), luteolin (Lut) and L-ascorbic acid (A.A.) from the literature. The uncertainties on the log values represent σ.

| | Method | Medium | °C | log $K^*$ | p$K_{a1}$ | p$K_{a2}$ | p$K_{a3}$ | p$K_{a4}$ | p$K_{a5}$ | Ref. |
|---|---|---|---|---|---|---|---|---|---|---|
| **Que** | Electroph. | $H_2O$ | 25 | | $7.1 \pm 0.1$ | $9.1 \pm 0.1$ | $11.1 \pm 0.4$ | | | [42] |
| | Colorim. Spectr. | $H_2O$ | 25 | $1.8 \pm 0.1$ | $6.4 \pm 0.1$ $6.6 \pm 0.1$ | $8.1 \pm 0.1$ $8.1 \pm 0.1$ | $9.0 \pm 0.1$ | $9.6 \pm 0.1$ | $11.3 \pm 0.1$ $11.4 \pm 0.1$ | [31] |
| **Lut** | Pot. | 0.16 M NaCl | 37 | | $8.9 \pm 0.1$ | | | | | [14] |
| **A.A.** | Cond. | $H_2O$ | 25 | | 4.147 | | | | | [45] |
| | Pot. | 0.16 M NaCl | 37 | | $3.86 \pm 0.01$ | | | | | [11] |
| | Spectr. | $H_2O$ | 25 | | $4.16 \pm 0.01$ | $11.73 \pm 0.02$ | | | | [46] |
| | Titr. | $H_2O$ | 16–18 | | 4.14 | 11.43 | | | | [47] |

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
