# Peer review of "Wide pH Range Potentiometric and Spectrophotometric Investigation into the Acidic Constants of Quercetin, Luteolin and l-Ascorbic Acid in Aqueous Media"

_applsci, doi:10.3390/app13020776_

Round 1
Reviewer 1 Report
Manuscript ID: applsci-2089071
Title: Wide pH range potentiometric and spectrophotometric investigation on the acidic constants of quercetin, luteolin and L-ascorbic acid in aqueous media
Journal: Applied Sciences
The manuscript is well structured, the discussion well developed, and the bibliography complete. In my opinion the research work is overall well done and the manuscript suitable for publication in Applied Sciences after very minor revision.
I have only a few minor corrections/additions to report:
1. Figure 2, add in the caption the meaning of the symbols.
2. Figure 3, add the experimental conditions (i.e. concentration, temperature and ionic strength).
3. Page 8, line 309 "aqueos" should be corrected in "aqueous".
Author Response
We thank the reviewer for his/her positive general comment and suggestions.

Reviewer 2 Report
Manuscript: Wide pH range potentiometric and spectrophotometric investigation on the acidic constants of quercetin, luteolin and L-ascorbic acid in aqueous media
Authors: Luana Malacaria and Emilia Furia
The manuscript reports a study on the acidic constants of quercetin, luteolin and L-ascorbic acid. The accurate study of the acid-base properties of a ligand is of considerable importance since numerous properties of the ligand depend on them: the chelating capacity towards metal ions, the antioxidant activities, the toxicity, etc.
The study is well conducted, the experimental procedure accurately described, and an in depth analysis of the data is reported. Furthermore, the use of two different instrumental techniques, potentiometry and spectrophotometry, makes it possible to analyze a wide range of pH and to supply particularly accurate constant values.
I think the manuscript can be accepted for publication on Applied Sciences and it needs only few minor revisions:
1. Paragraph 2.2. The authors report the concentrations of the ligand used for the potentiometric measurements but not those used for the spectrophotometric measurements. The latter must be added in the experimental part.
2. Caption of Figure 1. Explain to which experimental conditions the various symbols correspond
Author Response

(The authors gave the same response as above.)

Reviewer 3 Report
This contribution would have been considered as novel science forty years ago. Now, although the findings should be published, it is overlong and verbose in style; in parts it reads as if taken verbatim from a thesis. It should be shortened considerably, with structural and theoretical information being placed in an Appendix
Author Response

(The authors gave the same response as above.)

Round 2
Reviewer 3 Report
The authors have paid no attention to the comments of the referee and have made minimal alterations. I now, with regret, recommend rejection
Author Response
I apologize but in the previous revision I did not fully understand the meaning of reviewer’s comments.
So now, as requested in the first round, in the current revised version of my manuscript I have moved some of structural and theoretical information in the Appendix A.
